# Performing Arts in Suicide Prevention Strategies: A Scoping Review

**DOI:** 10.3390/ijerph192214948

**Published:** 2022-11-13

**Authors:** Chiara Davico, Alessandra Rossi Ghiglione, Elena Lonardelli, Francesca Di Franco, Federica Ricci, Daniele Marcotulli, Federica Graziano, Tatiana Begotti, Federico Amianto, Emanuela Calandri, Simona Tirocchi, Edoardo Giovanni Carlotti, Massimo Lenzi, Benedetto Vitiello, Marianna Mazza, Emanuele Caroppo

**Affiliations:** 1Section of Child and Adolescent Neuropsychiatry, Department of Public Health and Pediatric Sciences, University of Turin, 10100 Turin, Italy; 2Department of Humanities, Social Communities Theatre Centre, University of Turin, 10100 Turin, Italy; 3Department of Psychology, University of Turin, 10100 Turin, Italy; 4Department of Neuroscience, University of Turin, 10100 Turin, Italy; 5Department of Philosophy and Education Sciences, University of Turin, 10100 Turin, Italy; 6Department of Humanities, University of Turin, 10100 Turin, Italy; 7Institute of Psychiatry and Psychology, Department of Geriatrics, Neuroscience and Orthopedics, Fondazione Policlinico Universitario A. Gemelli IRCCS, Università Cattolica del Sacro Cuore, 00168 Rome, Italy; 8Department of Mental Health, Local Health Authority Roma 2, 00159 Rome, Italy

**Keywords:** suicide, prevention, performing arts, theater, role playing, gatekeepers

## Abstract

Suicide is a leading cause of death all over the world. Suicide prevention is possible and should be pursued through a variety of strategies. The importance of the arts for positive health outcomes has been increasingly evidenced. This scoping review aimed to identify the possible role of the performing arts—defined as a type of art performed through actions such as music, dance, or drama executed alive by an artist or other participant in the presence of an audience,—in suicide prevention programs. PubMed, Embase, PsycINFO, CINAHL, ProQuest Psychology Database, Scopus, and Web of Science were searched using terms in English for publications of original studies that included performing arts in suicide prevention programs. Thirty-five studies conducted between 1981 and 2021 were identified, of which only five were randomized clinical trials and four quasi-randomized studies. Interventions used different performing arts to improve awareness, self-efficacy, and soft skills relevant to suicide prevention. Studies were addressed mainly to gatekeepers but also directly to at-risk populations. While the study designs do not allow inferences to be drawn about the effectiveness of performing arts in preventing suicide, the review found that performing arts have been successfully implemented in suicide prevention programs. Research to evaluate the possible therapeutic benefit is warranted.

## 1. Introduction

### 1.1. Suicide and Its Prevention

Suicide is a leading cause of death all over the world, with an estimated death toll of about 700,000 per year. Globally, for all ages, the suicide rate is greater in males than females. More than half of suicides occur under 50 years of age. Between 15 and 29 years, suicide represents the top third cause of death for females and the top fourth for males [1]. Recent findings reveal that the incidence of suicide is rapidly increasing throughout adolescence [2,3]. Because suicide is a public health priority, the WHO member States are working to achieve the goal of reducing the global suicide rate by one-third by 2030.

Suicidal behavior has multiple causes that are broadly divided into proximal stressors or triggers and predispositions [4,5]. In our experience, suicide prevention must be seen as a set of actions that can, in various cases, reduce the risk of suicide, although it is never completely nullified precisely because of the multifactorial nature of suicide. Psychiatric illness is a major contributing factor, and mood disorders, especially major depressive disorder and bipolar disorder, are associated with about 60% of suicides [6]. It is known, however, that factors such as chronic pain or illness, legal, criminal, or financial problems, impulsive or aggressive tendencies, substance abuse, adverse childhood experiences, loss, discrimination, bullying or violence, social isolation, and stigma are all linked to an increased risk of attempting suicide [7].

Suicide prevention is possible [8,9] and should be pursued through different strategies. WHO recognizes that evidence-based interventions for suicide prevention should be organized in a framework that distinguishes between universal, selective, and indicated interventions in order to modify health systems, societies, communities, and individual relationships with a “comprehensive, multisectoral approach” [10,11].

Since no single method is clearly superior to others [12], suicide prevention should involve public health strategies at a national level by promoting public education campaigns to reduce stigmatization, limiting access to lethal methods, such as firearms, dangerous medicines, or pesticides, and educating the media for a responsible report of suicide news [13]. It is crucial to improve the awareness of suicide risk factors and depression management by primary care physicians and non-psychiatrist medical specialists.

At the same time, it is also very important to enhance soft skills and education in recognizing the warning signs in college and high schools among teachers, students, and gatekeepers in general. Contact with potentially vulnerable populations provides an opportunity to identify individuals with risk factors, suicidal ideation, or previous suicide attempts and to guide them in a facilitated way toward the appropriate assessment and treatment [9]. Gatekeeper training programs are needed for as many individuals as possible who have suicidal ideation but do not seek help and need to be “helped in seeking help” since suicide risk factors are recognizable and thus identifiable [14].

Gatekeepers are defined as “individuals in a community who have face-to-face contact with large numbers of community members as part of their usual routine” and, therefore, who may be trained to identify at-risk persons and refer them to the appropriate support services [15]. The category includes clergy, first responders, pharmacists, geriatric caregivers, staff, and those employed in institutional settings, such as schools, prisons, and the military.

At the same time, educating youths about mental health, in terms of awareness, coping skills, and self-referral, is another key element in the complex scenario of suicide prevention strategies [11,16,17,18].

Although one of the main goals of suicide prevention is to reduce mental illness stigmatization, it is important to consider that raising public awareness of suicide may have the unintended consequence of adversely affecting vulnerable individuals by suggesting that suicidal behaviors are somewhat widespread and can be an acceptable way to cope with difficulties. Thus, suicide prevention strategies must take into account both the potential benefits and possible risks of the interventions [19].

### 1.2. Performing Arts and Mental Health

Art is broadly defined as any means of expressing individual and social values through concrete and creative activities and processes [20]. According to Dewey’s conceptualization of art, artistic interventions can communicate a moral or educational purpose or explain experiences of the everyday emotional and rational inner world [21]. Art is a specific praxis focused on giving light and enabling the relationship between human beings and the world in which they live [22].

Although boundaries are sometimes difficult to define, the arts can roughly be divided into performing arts, visual arts, literature, culture, and digital arts. In recent years, there has been an increasing interest in research into the effects of the arts on health. The arts, in fact, strengthen a relationship with cultural identity, and their use can have positive effects on mental and physical well-being by promoting self-understanding, expression, confidence, self-esteem, and both verbal and nonverbal communication [23,24,25,26].

The arts can also contribute to the management and treatment of illness across a lifespan, as underlined by a recent scoping review by the WHO [27]. Performing arts provide imaginative experiences for both the art producer and the audience and provoke an emotional response and cognitive stimulation as well as stress reduction; the creative process of art production, in fact, requires novelty, creativity, and originality together with specialized and technical skills [27].

Since the mid-20th century, especially in Western countries, there has been a great deal of interest in evaluating the effect of the arts on health. In particular, the use of performing arts, defined as types of art performed through actions executed live by the artist or other participants for a present audience, such as music, dance, or drama, is becoming widely used as a tool for physical and especially mental health prevention, as well as promotion and treatment in care settings, as recently reviewed by Gaiha and colleagues [28]. In Italy, for example, one area of new theatrical engagement has been in psychiatric residential settings, just at the time when mental health policy reforms mandated by the 1978 Basaglia law led to the deinstitutionalization of psychiatric patients [29].

In recent years, there has been an increasing use of the arts to counter the stigma that still exists against mental illness. Interactive theater, for example, has been often recognized as a tool for promoting prosocial behavior and addressing deeply held stigmas [30]. This kind of prejudice is also being challenged by demonstrating the role of performing arts in promoting reintegration into employment, facilitating skill development, and the ability to engage in learned behaviors and breaking down barriers between people with and without mental illness [27,31,32,33].

In many cases, the experiences based on performing arts are developed for groups of people sharing the same conditions—such as youth or ill and vulnerable people—or the same cultural background—in particular, people living in rural areas, immigrants, or people affected by conflicts— [27,34,35,36]. An interactive performance model, in which students actively engaged in their own learning through dialogue, experimentation, and movement, seemed to be more effective than a traditional classroom approach in increasing students’ willingness to comfort victims of sexual violence in distress [37]. People participate in the creation of artwork under the guidance of specialists: for this reason, it is crucial for these people to be trained to administer this type of intervention. It is also important for trainers to be aware of the possible dangers of these activities [38,39] in terms of being engaged in “demanding” activities that could have psychological negative effects on patients [40].

Performing art forms have demonstrated improvements in physiological parameters, such as blood pressure, heart rate, and immune status in patients with cancer, respiratory or cardiac disease, or diabetes, and, in the case of cancer patients, there was a reduction in anxiety and depression [27]. Dance and movements offered patients with mental illnesses a good way to communicate and increase physical activity, and in patients with Parkinson’s disease, they reduced the sense of isolation [27,41]. According to some studies, involvement in performing arts reduced the risk of developing depression in adolescence or later in life, as active participation helped improve self-confidence. Children and adolescents who participated in art programs reported increased communication skills, anger management, and higher levels of well-being, socialization, and resilience than at the beginning of the experience [42,43]. Other studies indicated an improvement in life skills, coping skills, prosocial attitudes and behaviors, an increase in school performance, and the confidence to help their peers in ways not sufficiently supported by traditional programs [44,45,46]. Art classes involving creative writing, dancing, or listening to music strengthened the sense of mutual support among health care workers and improved the work environment by decreasing conflicts. Art programs were also found to facilitate caregiver–care interaction and intensify emotional responses, thus making treatments more effective [27,47].

Different art forms can also be helpful for the caregivers themselves, who can, thus, listen to their own needs, reduce stressors, increasingly improve empathy, enjoy moments of catharsis, and build a positive image of themselves and their effectiveness [27].

Much attention has also been given to the inclusion of performing arts, for example through role-playing, in the education and training of mental health workers, especially when considering the potential effects of these interventions on communication skills, empathy, and understanding of the needs of patients. In most cases, health care workers receiving this type of education understood how to respond more humanely to medical, ethical, and social needs and reported greater satisfaction with their jobs [48]. Participants could either be exposed to the arts (e.g., as active observers) or create their own arts (e.g., actors or writing dramaturgy). In both situations, they are active and involved in neuronal cognitive interactions with the play [49] and in dynamic present interactions with the trainers through, for example, the application of role-playing skills in a variety of hypothetical contexts and situations [27,50].

These approaches appear to be superior to passive techniques; the use of passive training methods, in fact, has been shown to have diminutive effects on both skill acquisition and subsequent behavior, in large parts, due to the limited opportunity to practice and receive constructive feedback on skill usage [51,52]. With active methods, it is easier to implement new skills, and participants can control their skill development (e.g., by practicing learned skills, asking questions, and receiving feedback appropriate to their performance) through dynamic interactions with trainers [31,33,53].

Performing arts that exploit these modalities can offer truly active cultural participation by involving participants in the creative process at any level through the assignment of active roles, such as that of actors, authors, organizers, or members of an active audience. It is also worth noting how the “real-time” dimension of the performing arts requires a specific social and physical encounter between participants, both in the creation of the product and in its fruition.

For the purposes of this review, performing arts include participatory theater, dramatic improvisation, role-playing, and in person attendance of a theater performance.

### 1.3. Performing Arts in Suicide Prevention

The creative expression that accompanies the art process is an engaging emotional experience that can powerfully contrast with the implicit destructiveness of suicide behaviors [29,54]. As mentioned above, there is clear evidence that art interventions can support mental health [27,55,56,57,58] since they facilitate dialogue, reduce stigma, and enhance expression, coping skills, empathy, and personal and cultural resonance, all of which address risk factors for suicide [27,59,60]. They can also facilitate the expression of emotions such as entrapment, loneliness, and burdensomeness [61], enhance belonging [57], and protect against suicidal ideation [62]. As with alcohol prevention and other public health topics, research on suicide prevention shows that didactic education and awareness-raising alone will not reduce the risk [63,64]. Instead, more direct and personal forms of involvement are needed [34,65].

Indeed, a major limitation of suicide prevention programs is the reliance on passive training techniques, such as listening to in-person lectures and self-study (e.g., online readings and watching videos) [66]. Hence, active learning techniques should be implemented, as they can help facilitate skill development and the ability to engage in learned behaviors [31,32,33,53]. As mentioned above, active learning techniques consist of dynamic interactions between the participants and the trainer through the application of skills in role-playing in a variety of contexts and hypothetical situations. These approaches appear to be superior to passive approaches in that participants can take control of their own activity.

Using theater programs as a suicide-prevention tool is not new. In 1974, Jackson and Potkay [67] reported favorable reactions from college-aged audiences to an educational play regarding suicide. More generally, the participatory theater has long been recognized as a tool for promoting prosocial behavior and addressing deeply held stigmas [34,37,68]. Seeing coping skills modeled through performance or direct interpersonal contact may give students the confidence needed to help their peers in ways not sufficiently supported by the traditional curricula [46]. As reported by Wasserman et al. [65], interactive role-playing with students can influence both suicide attempts and help-seeking behaviors.

### 1.4. Aims of This Review

Despite the potential value that the performing arts can have for suicide prevention, little systematic assessment of their utilization and therapeutic effects has been reported. We conducted a scoping review with the aim of evaluating the currently available evidence of the possible applications of performing arts to suicide prevention. Since we expected to find a very heterogeneous set of publications, we chose the format of the scoping review in order to provide a wide overview of what is currently known on the topic. More specifically, the review aimed at addressing the following main questions: (1) *Study design*: Which study designs have been used to assess the feasibility, benefit, and possible adverse effects? On which populations? In which settings? (2) *Purpose and outcomes*: Were there any attempts to estimate effectiveness? Which outcomes have been examined? What is the evidence for feasibility and effectiveness? (3) *Art Forms*: What types of performing arts have been used in suicide prevention?

## 2. Materials and Methods

The Joanna Briggs Institute (JBI) methodology for scoping reviews, described in the online JBI Reviewer’s Manual [69], was employed for this review. The results are presented following the preferred reporting items for systematic reviews and meta-analyses extension for scoping reviews (PRISMA-ScR) checklist [70]. No a priori protocol was registered. Further information on the process can be obtained from the corresponding author upon request.


*Search Strategy*


The review covers data published between 1981 and 2021. Selected keywords were combined to create search strategies which were adjusted for each screened database. Articles were searched in the following databases: PubMed, Embase, CINAHL, PsycINFO, Scopus ProQuest Psychology Database, and Web of Science on 25 May 2022. We searched for articles reporting the use of performing arts (theat* OR drama* OR recit* OR “performing-art*” OR “participatory-art*” OR psychodram* OR “role-play” OR “role-plays” OR “role-playing” OR “role-player*” OR roleplay* OR impersonat* OR storytell* OR “story-tell*”) as a suicide prevention strategy.

The complete search string is available in Table 1. Hand-searching of the gray literature sources was also conducted, and additional references identified through other sources with a title and abstract (Google Scholar, Cochrane Library) were included for consideration.

### 2.1. Inclusion and Exclusion Criteria

The inclusion criteria to select the articles for this review were based on the population, concept, and context (PCC) elements reported below.

Population: Interventions could directly target specific at-high-risk populations or indirectly target gatekeepers. As regards the age range, we included articles considering both adults and youth.

Concept: We searched for original articles reporting about interventions using the performing arts as a suicide prevention strategy. We included participatory theater, dramatic improvisation, role-playing, role-playing online simulation, the vision of theater performance, and education-like applied theater as types of performing arts. Drama therapy has been excluded from this review since it is a specific treatment approach with a clear-cut therapeutic goal. We also excluded cinema and, in general, watching movies or participating in a cinema club because these imply a passive attitude, while theater performances involve audience participation in a more direct way since the performers and audience interact in a constant cyclic interchange. Moreover, each play or performance is an experience that exists in a finite space and time and has a sort of uniqueness, being much more linked to the relationship between that particular performer and the attending audience. We excluded book chapters or sections, reviews, thesis, conference proceedings or conference papers, and poster contributions to scientific congresses. We excluded all original papers that were not in English and all articles we were not able to find.

Context: The approach was inclusive, and no cultural, geographical, race, or gender-specific limits were set.

### 2.2. Screening and Selection of Articles

Articles were initially screened based on their titles and abstracts according to the criteria previously described. Duplicates were removed. Full-text papers published in English in peer-reviewed journals since 1981 were selected. The first 50 articles were examined together by two authors (EL and FDF); then, 10% of all the articles were independently screened for eligibility by the two authors, and a consensus was reached through discussion between the two authors and consultation with a third author (CD); finally, all the remaining articles were evaluated for their eligibility independently by the two reviewers and, in case of discordant opinion, consensus was reached with input from the third reviewer (CD).

### 2.3. Extraction and Presentation of Results

All the data relevant to inform the scoping review objectives and questions were extracted from the articles meeting the inclusion criteria. The strength of the evidence for each article was assessed according to the levels of evidence developed by the JBI [69].

The results were grouped according to the year, country, study design, main art forms (theater performance, role-playing), purpose, sample size, study population, and period of intervention. The study design has been classified according to the JBI levels of evidence [71].

## 3. Results

### 3.1. Selected Publications

The search yielded a total of 5737 records (Figure 1). After electronically eliminating duplicates, the titles and abstracts of the remaining 3086 records were manually screened for suitability based on the stated inclusion and exclusion criteria. Based on this review, another 36 duplicate publications were identified and, therefore, discarded. Among the remaining records, 3015 were excluded since they did not meet the criteria. A total of 35 publications were eventually included in the scoping review (Table 2).

### 3.2. Study Design—Which Study Designs Have Been Used to Assess Feasibility, Benefit, and Possible Adverse Effects? on Which Populations and Settings? What Is the Evidence for Feasibility and Effectiveness?

The selected studies are characterized by their extreme heterogeneity in terms of study design, type and size of populations studied, their purpose, and outcomes. Of the 35 included studies, most of them were observational studies without a control group (*n* = 22) [30,50,72,73,74,75,76,77,78,79,80,81,82,83,84,85,86,87,88,89,90,91], four were quasi-experimental prospective controlled studies [92,93,94,95], four were cross-sectional studies [96,97,98,99] and five were randomized control trials (RCTs) [18,100,101,102,103].

Most of the studies were published from 2010 onward (29, 83%) [18,30,50,64,76,77,78,79,80,81,82,83,85,86,87,88,89,90,91,93,94,95,97,98,99,100,101,102,103]. Regarding geographical provenience, 54% of the studies were conducted in the US (19) [30,50,74,75,76,82,83,84,86,87,88,89,90,93,94,99,100,101,103], four in the UK [72,78,92,96], three in Australia [81,91,102] and two in Canada [77,80]. The remaining studies were from Norway (one) [73], Russia (one) [97], Ireland (one) [79], Japan (one) [85], and India (one) [95]. Two studies reported a European experience (the Saving and Empowering Young Lives in Europe–SEYLE- and Youth Aware of Mental Health Program–YAM-study) and had a mixed population, respectively, from the Austrian, Estonia, France, Germany, Hungary, Ireland, Italy, Romania, Slovenia, and Spain–SEYLE study [18], and from the Estonian, Italy, Romanian and Spain–YAM study [98].

Most of the studies (22) [18,50,74,75,76,79,82,83,84,85,86,87,90,91,92,93,94,95,96,100,101,103] used quantitative methods to assess their outcomes, while a smaller portion used a qualitative approach (7) [30,73,77,88,98,99,102] and six studies used mixed methods, with both qualitative and quantitative data collection or analysis techniques [72,78,80,81,89,97].

The studies were heterogeneous as to the sample size: fourteen studies (40%) had a very small sample size (from 10 to 50 participants), eight studies had a sample size from 51 to 100, five studies had a medium sample size (ranging from 101 to 349 subjects), and eight studies had a very large sample size of up to 33,703 participants [86]. The studies with the largest sample sizes (9000, 11,110, 18,896, and 33,703 participants [18,82,86,101], used online role-playing simulator techniques, except for the SEYLE study [18].

Regarding the participants, most of the studies were addressed to gatekeepers as school staff members, educators, university professors, college or high school students, social workers, and nonclinical employers in hospitals (twelve studies, 31.4%) [74,76,81,82,86,87,90,92,93,94,100,103]; nine studies were addressed to youth, adolescents, college or high school students as vulnerable and high-risk populations [18,30,75,80,83,95,96,98,99], ten to health professionals in training [50,72,73,78,79,84,85,88,89,102], such as medical school students, nurse practitioners, residents, physicians, and pharmacy staff members. Two studies were addressed to the adult population [77,91], and finally, three had mixed recipients (i.e., schoolteachers and mental health professionals or gatekeepers and adolescents, as in the SEYLE study) [18,97,101].

### 3.3. Purpose and Outcomes—Was There Any Attempt to Estimate Effectiveness? Which Outcomes Have Been Examined? What Is the Evidence for Feasibility and Effectiveness?

Four studies reported on the outcomes of online gatekeeper training programs, with positive outcomes in terms of preparedness, likelihood, and self-efficacy [82,86,101]. In Vallance’s study, attitudes towards a novel application of learning technology were also evaluated in terms of usability, utility, and improvements in psychiatric skills/knowledge. Users also expressed less anxiety and more enjoyment than when role-playing face-to-face [78].

Fourteen studies reported on the outcomes of different gatekeeper training programs in which role-playing was used [50,72,74,76,79,81,85,87,88,89,90,93,97,102]. The studies were mainly focused on evaluating pre-post knowledge and perceived skills (attitudes), but some studies also analyzed the diffusion of knowledge, satisfaction, behavior, and the acceptability of the program. Some studies also planned a follow-up evaluation of the post-intervention evaluated competencies [81,85,90].

Four studies, two RCTs [100,103] and two quasi-experimental prospective controlled studies [92,94], compared different gatekeeper training programs. In particular, Coleman tested hypotheses about two types of brief suicide prevention gatekeeper training (question, persuade, refer [QPR] and RESPONSE) and one longer suicide intervention skills training (Applied Suicide Intervention Skills Training [ASIST]), finding that all three types of training showed large changes in prevention attitudes and self-efficacy that was largely maintained at follow-up. ASIST trainees showed large increases in asking at-risk youth about suicide at follow-up [94]. Cross et al. compared gatekeeper training as usual with training plus brief behavioral rehearsal (i.e., role-playing). They found that both training conditions resulted in enhanced knowledge and attitudes, but interestingly, behavioral rehearsal with role-playing practice resulted in higher total gatekeeper skill scores both immediately after training and at follow-up [100]. Godoy and colleagues aimed to examine the impact of two training enhancements (role-playing and booster training) to QPR gatekeeper training programs and found that at the six-month follow-up, among the participants assigned to role-playing, a significantly larger proportion of those were assigned booster performed identifications and referrals of high-risk subjects [103]. Finally, Fenwick et al. compared the impact of two types of training courses: full-day workshops with actors using role-playing with patients and a half-day lecture and found that both types, of course, led to improvements in skills and confidence, which were sustained at a two-month follow up [92].

Two other studies were focused on reducing the stigma and isolation surrounding a patient’s suicide [84] and described the experience of dealing with suicidal patients [73] with a qualitative approach, providing additional information about this delicate theme.

A description of interactive approaches which increased awareness of mental health issues was provided both by Wasserman, who interviewed youths who participated in the YAM program, a universal mental health promotion program [98] and by Fanian, who described art model interventions for youths in North-West Territories in Canada, where they explored critical community issues and found solutions together using the performing arts [80]. Different studies focused on increasing awareness of mental health and suicide problems or an increase in the willingness to seek help or to struggle against stigma [30,83,91,96], for example, with positive outcomes in terms of self-efficacy for communicating about suicidal thoughts or seeking help [30,83]. Keller and colleagues analyzed differences in Eastern Montana Caucasian and Native American adolescents and young adults’ experiences with stigma about mental illness that affected help-seeking for suicidal experiences. Using textual analysis, they found that for both ethnic groups, stigma is a barrier to expressing emotional vulnerability, seeking help, and acknowledging mental illness [99].

Two studies were aimed at reducing suicidal behaviors among adolescents. An Indian quasi-experimental prospective controlled study [95] tested the effect of life skills training on suicidal behavior in 950 adolescents in the ninth grade. They measured the frequency of suicidal behavior pre-and post-intervention with a self-report psychometric test and found a significant effect of life skills training in reducing suicidal behavior. The training also improved decision-making, problem-solving, goal-setting, conflict resolution, advocacy, coping, and mindfulness skills for students. The SEYLE study was a multicenter, cluster-randomized controlled trial. Participating schools were randomly assigned to receive one of three interventions or as a control group. QPR is a gatekeeper training module targeting teachers and other school personnel, the YAM targeting pupils, and screening by professionals (ProfScreen) with the referral of at-risk pupils. The primary outcome measure was the number of suicide attempt(s) made by a 3-month and 12-month follow-up. No significant differences between the intervention and control group were recorded at the 3-month follow-up however, at the 12-month follow-up, YAM was associated with a statistically significant reduction in incident suicide attempts and severe suicidal ideation compared with the control group [18].

Stewart and Colleagues examined the effect of the Prodigy Cultural Arts Program on at-risk and adjudicated youths in a rural and an urban locale, with specific outcomes on mental health symptoms, delinquency, and family functioning. The results suggest a significant improvement in family functioning overall as well as statistically significant changes in mental health symptoms, including depression/anxiety, somatic, and suicidal symptoms for both males and females and in both urban and rural settings. The authors also underlined that females appeared to especially benefit from the program [75]. A two-day art-based symposium that brought together members from diverse cultural communities has been described in a qualitative way by Silverman et al., with the objective of gathering information on the participants’ experiences of exploring the issue of suicide within an art-based approach and of determining if cross-cultural themes would emerge. They found that using the arts helped to facilitate dialogue and communication, and that specific cross-cultural themes did emerge [77].

Only two studies reported detailed information on costs and feasibility: SAVE [88] was chosen over six other online gatekeeper programs because it was available free of charge and is relatively short. Owen’s study reports detailed each item’s costs (venue, food, support for statistical analysis, faculty members’ time). All the other studies did not report cost details or other information on the feasibility of their interventions [84]

Twenty-five interventions are specific to suicide [18,30,50,72,73,74,76,77,78,81,83,84,85,87,88,89,90,92,93,94,95,99,100,102,103], and the remaining 10 include suicide but also broaden to include mental health [75,80,86,91,97,98,101], stigma [96], psychological distress [82], and substance abuse [79].

### 3.4. Art Forms—Which Type of Performing Arts Have Been Used on Suicide Prevention?

The art forms used in the studies and included in the review were grouped into three forms: theater, role-playing, and multiple art forms.

Twenty-seven studies used primarily role-playing [18,50,72,73,74,76,78,79,81,82,84,85,86,87,88,89,90,92,93,94,95,97,98,100,101,102,103], five studies involved mainly theater [30,83,91,96,99], three studies involved multiple art forms [75,77,80], as speech, sound production, and design, film, photography, multimedia arts, jewelry making, visual and performing arts, music and theater, artistic creation and writing).

The components of the art forms used in the studies were articulated as follows.

#### 3.4.1. Theater

Keller and Wilkinson [83] and Keller et al. [30] focused on *Let us Talk*, a community-based suicide prevention performance that was created by community and university theater directors in collaboration with a health research team and student performers. Keller et al. [99] analyzed performances that originated from five theater workshops led by professional theater directors and groups of volunteers. During the playwriting process, the student-actors shared memories, songs, and poems with writer-actors to develop a creative script based on their own experiences of depression, suicidal ideation, suicide attempts, and grief over the suicide of a friend or family member in meetings of 2–3 h per week for 10 weeks. Nash’s study [91] evaluated a workshop for health students that used filmed vignettes of a verbatim theater. The STIGMA program [96], provided to students in the Scottish Highlands, consisted of a play with professional actors, a workshop, and evaluation forms to address sensitive issues close to young people, such as suicide or self-harm, and to identify individuals who were struggling or discouraged from seeking help. The duration of the intervention (the play and workshop) was one hour and ten minutes, and information booklets were distributed at the end.

#### 3.4.2. Role Playing

The suicide prevention gatekeeper training program QPR (question, persuade, refer) [104] was used in six studies [18,74,76,94,100,103]. It consisted of a lecture, a 10 min introductory video, the distribution of summary handouts and reference cards, and a question-and-answer discussion period. The participants were given an additional opportunity to use role-playing in small groups. Each group was given a scenario that included a precipitating problem, multiple suicide warning signs, and indications of the need for intervention where they had to play the roles of the suicidal student and the adult caretaker.

The YAM program was developed for the SEYLE study [18] as a coded intervention aimed at all students to promote a greater understanding of mental health, involving 3 h of role-playing sessions with interactive workshops. YAM aimed to increase mental health awareness of the risks and protective factors associated with suicide, including knowledge of depression and anxiety, and to improve skills for coping with adverse life events, stress, and suicidal behavior. YAM was also examined in Wasserman’s study [98], in which researchers investigated youth’s levels of motivation, their ease with engaging in dialogue with mental health professionals, and comfort with the format and content of YAM through interviews in a qualitative way.

The Campus Connect Gatekeeper course applied in Pasco’s study [93] is part of the standard orientation for resident advisors at Syracuse University. During their training, participants are provided with information on suicide prevalence rates among college students, suicide warning signs, and strategies for asking students if they are thinking about suicide. Additionally, participants are instructed on active listening skills and are guided through multiple experiential exercises designed to provide practice, ask questions about suicide, and practice active listening skills. All training courses conclude with role-playing.

Kratz’s study [87] explored the educational outcomes of a four-week course on suicidology. The course was characterized by a multimodal structure, which also included role-playing, in which each student spent 50 min with a standardized client presenting a moderate to a severe suicidal crisis.

Some authors organized day-long workshops [85,92] with actors playing depressed patients. A similar strategy was provided in a training course for mental health practitioners [50] in which, after the use of role-playing, trainers and participants discussed how to formulate risks and develop a safety net for the client at risk in the exemplary case. A specific type of high-fidelity simulation was the objective structured clinical examination (OSCE) [89], a high-quality, standardized experience designed to enable nursing students to master skills in the clinical setting. Students encountered a patient who was presented to a mental health clinic experiencing specific psychiatric symptoms in order to build the clinical skills needed to provide pediatric psychiatric care.

Nine other studies [72,73,79,81,84,90,95,97,102] used role-playing in suicide prevention interventions and training programs for gatekeepers, but no detailed information on role-playing was given in the text.

Different studies included in this review used online role-playing. The *At-Risk for Elementary School Educators* simulation is an online digital experience developed by Kognito and used in two studies [86,101]. It is a self-directed online simulation that takes 45 to 90 min. The simulation platform offers role-playing experiences with a virtual student and parent who have similar emotions, personalities, memories, and reactions to students experiencing psychological distress and their parents.

Other online simulations have been used [82], in which the participant interacts with computer-guided avatars or virtual humans rather than another person in highly replicable virtual role-playing games. Users interact with avatars of intelligent, fully animated, and emotionally responsive students who are experiencing psychological distress, such as suicidal thoughts. Additionally, in the study of Vallance [78], users interacted only through their avatars, communicating via audio-microphone headsets. The role-playing phase involved students using their avatars to interact with the teacher, playing the avatar of a “depressed teenager”.

The online adaptation of the Veteran Administration’s suicide prevention gatekeeper training program (SAVE) was developed for community pharmacy staff [88]. It is a relatively short online gatekeeper program that models telephone and in-person interactions with at-risk patients using semi-structured interviews.

### 3.5. Multiple Art Forms

The Kǫts’iìhtła (‘Let us light the fire’) project was a five-day arts and music workshop for young people that took place in the community of Behchokǫ, North-West Territories, Canada, with the aim of empowering young people to explore critical issues facing their community and their lives and to find solutions together using the arts [80].

Silverman [77] led a two-day art-based symposium in which, through drama, music, artmaking and writing, a group of members from diverse cultural communities was able to share complex feelings and share their thoughts in a creative way through drama, music, artmaking, and writing in order to explore the topic of suicide from their own cultural perspectives. Mixed arts, such as visual, performing, musical, media, and theatrical arts, have also been used in the Prodigy Cultural Arts Program [75] addressed to young people who had been adjudicated in the juvenile justice system and at-risk non-offending youth in the community.

The classes (requiring three hours per week for eight weeks) were taught by master artists from the community who deliberately developed positive and supportive relationships with the young people.

## 4. Discussion

This scoping review found indications supporting the use of performing arts in suicide prevention interventions: performing arts are useful to improve awareness, self-efficacy, and soft skills relevant to suicide prevention, both in gatekeeper training programs and in interventions with high-risk populations. However, given the extreme heterogeneity in terms of the study design, type and size of populations studied, purpose and outcomes, with a few RCTs and mainly observational studies without control groups and small sample sizes, no firm conclusions about effectiveness can be drawn. For most studies, the main goal was to describe a specific kind of intervention, mostly gatekeeper training programs, with a simple pre- and post-intervention evaluation, very often without any control group. A few studies reported evaluation at follow-up. Moreover, many studies presented a sample bias because of self-selection in that participants volunteered to participate, so generalization may be limited. Skills retention over time was evaluated only in a few studies.

The strongest results obtained from four of the five RCTs included in this review underlined the utility and advantage of role-playing techniques in gatekeeper training programs [18,94,100,103]. Coleman, Cross, and Godoy analyzed immediate outcomes, i.e., trainee attitudes and knowledge of the gatekeepers with superior results for the prevention programs, including role-playing. The SEYLE study demonstrated a significant reduction in incident suicide attempts and severe suicidal ideation (which is an ultimate goal of the gatekeeper training program as described by Coleman et al. [94]) at the 12-month follow-up for the YAM program, which involves role-playing sessions with interactive workshops for students [18], compared to the control.

Despite the methodological limitations, the identified studies reported that, overall, performing art techniques had a positive impact on suicide prevention programs. In fact, no studies reported negative outcomes or unintended harm. Thus, performing arts can be tentatively considered to be a useful tool to enhance the effects of suicide prevention programs. This was mainly shown by the studies of gatekeepers’ training. Fewer studies directly involved high-risk populations, probably because their inclusion in research is more challenging for logistic and ethical issues.

Data on the feasibility of implementation in community settings are rather limited. Very few studies [84,88] reported details on costs or other practical information. This is a significant limitation that should be addressed in future studies. When choosing to adopt a specific intervention program, public health policymakers must consider not only the effectiveness but also cost-effectiveness. Another point is that all the studies except one [95] were conducted in economically developed countries, underlining the need to also implement research on performing arts interventions for suicide prevention in disadvantaged settings and especially in those contexts where low-cost interventions, such as the performing arts, are needed to reduce stigma and enhance mental health awareness [35].

Keeping in mind that raising public awareness of suicide may adversely affect vulnerable individuals [19], we searched the identified studies for possible unintended harm from the preventive interventions, but none reported negative outcomes. A possible publication bias, with only positive outcomes studies being published, could be involved in this finding.

When planning this review, we expected to find more studies of theater, drama, or plays in suicide prevention programs. However, we found that most of the studies used role-playing as an active learning method for training gatekeepers. In many programs, role-playing provided the opportunity to conduct quasi-realistic encounters with “patients” in a controlled learning environment to improve competence in managing patients at risk of suicide. Indeed, role-playing allows learners to participate in realistic situations that might be stressful in real life, allowing them to make mistakes and learn in a stress-free environment that has been shown to facilitate skill acquisition [101].

Gatekeeper training was effective in all the studies in changing attitudes and building knowledge, but there are still little data on how it increases prevention behavior, that is, about the effect on a real-life context. Indeed, many interventions report mainly educational benefits and self-perceived behavioral changes rather than actual changes in clinical practice [97].

As mentioned before, research by Cross and colleagues [76,100] showed that adding role-playing to QPR achieved promising results. Cross et al. [100] found that the addition of role-playing led to an increase in almost half of the standard deviation in prevention behavior compared to standard QPR. Their role-playing is a 25 min addition to the QPR that begins with modeling desired prevention behaviors, including the suicide question, followed by practicing these behaviors in small groups: this seems to be a feasible and effective addition to the QPR or other short gatekeeper training courses.

As expected, theater-based narrative engagement approaches can support comprehensive suicide prevention programs by helping to identify and address entrenched beliefs and stigmas and empowering participants to learn about and access resources in their community [83]. The narrative format of the *Let us Talk* approach allows actors and audiences to acknowledge each other’s vulnerabilities and fears and collectively empower themselves [68]. Overall, in all the groups, participants’ perceptions of efficacy increased after the *Let us Talk* performances. The open discussion of loss, ideation, and suicide attempts presented in the live theater performance reduced audience members’ fears of contacting professional adults regarding suicide risk and was helpful in increasing students’ perceptions of their efficacy in contacting teachers or professionals regarding suicide risk.

Since we found very few studies using pure theater techniques, we hypothesized that, even though it is a very widespread method of psychosocial intervention, it remains too little studied and described in biomedical research, while more of the literature is available from a social sciences perspective, thus pointing to the need of trans-disciplinary research projects.

## 5. Limitations

The results of this review are limited by the small number of studies that used rigorous methodology to assess effectiveness.

Another limitation is the considerable heterogeneity of the studies with respect to design, population, and purpose. Moreover, the review included both studies using theater techniques and studies using role-playing in a more didactic way, which are at quite different ends of the “performing arts” concept, even though both require active participation. Even more distant from the participatory theater is online role-playing, which is anyhow included in this review since it requires interaction with participants and is an interesting and promising way to engage trainees in suicide prevention programs.

A more general limitation is that publication biases could not be assessed given the nature of the study outcome measures, so it is possible that studies reporting more favorable experiences with performing arts were more likely to be published.

## 6. Conclusions

The studies included in this scoping review provide overall support for the inclusion of performing arts in suicide prevention programs, both when these are used to train gatekeepers and when they are addressed directly to adolescents or high-risk populations. The more rigorous studies, with a randomized controlled design, demonstrated the clear effectiveness of these programs, including role-playing techniques, while other studies mostly indicated good responses in terms of increased knowledge, self-efficacy, and awareness, without strong evidence of efficacy. In role-playing, the component directly involving the trainee in actual performance seems to be a key factor in enhancing the learning process, while in the theater-based intervention, it is the emotional involvement that increases awareness and eventually facilitates help-seeking behaviors by reducing suicide-related stigma. However, the specific elements of performing arts that significantly contribute to the effectiveness of these interventions need to be further evaluated in future studies.

The small number of studies on the effects of performing arts, and especially theater techniques, on suicide prevention, is in marked contrast with the real-world diffusion of these forms of psychosocial interventions in schools, health care settings, residential psychiatric programs, and prisons. There is a need for closer collaboration between researchers in the medical field and those in art institutions. Further studies are warranted to explore the possible therapeutic benefit of performing arts in suicide prevention strategies.

## Figures and Tables

**Figure 1 ijerph-19-14948-f001:**
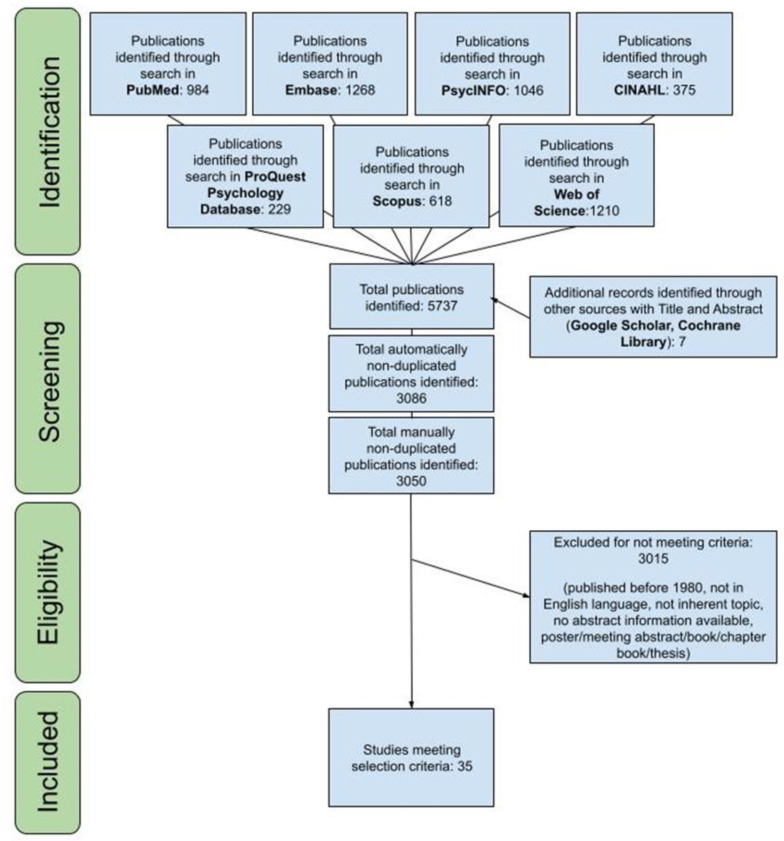
Study selection flow chart.

**Table 1 ijerph-19-14948-t001:** Complete search string.

Database	Search String
PubMed	(“Suicide”[Mesh] OR suicid*[tiab]) AND (“Drama”[Mesh] OR theat*[tiab] OR drama*[tiab] OR recit*[tiab] OR “performing-art*”[tiab] OR “participatory-art*”[tiab] OR “Psychodrama”[Mesh] OR psychodram*[tiab] OR “role-play”[tiab] OR “role-plays”[tiab] OR “role-playing”[tiab] OR “role-player*”[tiab] OR roleplay*[tiab] OR impersonat*[tiab] OR storytell*[tiab] OR “story-tell*”[tiab])
Embase	((‘suicidal behavior’/exp OR suicid*:ti,ab,kw) AND (‘drama therapy’/exp OR theat*:ti,ab,kw OR drama*:ti,ab,kw OR recit*:ti,ab,kw OR ‘performing arts’/exp OR ‘performing-art*’:ti,ab,kw OR ‘participatory-art*’:ti,ab,kw OR ‘psychodrama’/exp OR ‘role playing’/exp OR psychodram*:ti,ab,kw OR ‘role-play’:ti,ab,kw OR ‘role-plays’:ti,ab,kw OR ‘role-playing’:ti,ab,kw OR ‘role-player*’:ti,ab,kw OR roleplay*:ti,ab,kw OR impersonat*:ti,ab,kw OR ‘storytelling’/exp OR storytell*:ti,ab,kw OR ‘story-tell*’:ti,ab,kw)) NOT ‘conference abstract’/it
CINAHL	(MH “Suicide+” OR TI suicid* OR AB suicid*) AND (MH “Drama” OR TI theat* OR AB theat* OR TI drama* OR AB drama* OR TI recit* OR AB recit* OR MH “Performing Arts” OR TI “performing-art*” OR AB “performing-art*” OR TI “participatory-art*” OR AB “participatory-art*” OR MH “Psychodrama+” OR TI psychodram* OR AB psychodram* OR TI “role-play” OR AB “role-play” OR TI “role-plays” OR AB “role-plays” OR TI “role-playing” OR AB “role-playing” OR TI “role-player*” OR AB “role-player*” OR TI roleplay* OR AB roleplay* OR TI impersonat* OR AB impersonat* OR MH “Storytelling+” OR TI storytell* OR AB storytell* OR TI “story-tell*” OR AB “story-tell*”)
PsycINFO	(DE “Suicidal Behavior” OR DE “Attempted Suicide” OR DE “Suicidal Ideation” OR DE “Suicide” OR DE “Suicidality” OR DE “Suicide Prevention” OR TI suicid* OR AB suicid*) AND (DE “Theatre” OR DE “Drama” OR TI theat* OR AB theat* OR TI drama* OR AB drama* OR TI recit* OR AB recit* OR MH “Performing Arts” OR TI “performing-art*” OR AB “performing-art*” OR TI “participatory-art*” OR AB “participatory-art*” OR DE “Psychodrama” OR DE “Role Playing” OR TI psychodram* OR AB psychodram* OR TI “role-play” OR AB “role-play” OR TI “role-plays” OR AB “role-plays” OR TI “role-playing” OR AB “role-playing” OR TI “role-player*” OR AB “role-player*” OR TI roleplay* OR AB roleplay* OR TI impersonat* OR AB impersonat* OR DE “Storytelling” OR TI storytell* OR AB storytell* OR TI “story-tell*” OR AB “story-tell*”)
ProQuest Psychology Database	(MAINSUBJECT.EXACT(“Suicides & suicide attempts”) OR ti(suicid*) OR ab(suicid*)) MAINSUBJECT.EXACT(“Drama”) OR MAINSUBJECT.EXACT(“Acting”) OR MAINSUBJECT.EXACT(“Performing arts”) OR MAINSUBJECT. EXACT(“Role playing”) OR MAINSUBJECT.EXACT(“Storytelling”) ti(theat* OR drama* OR recit* OR “performing-art*” OR “participatory-art*” OR psychodram* OR “role-play” OR “role-plays” OR “role-playing” OR “role-player*” OR roleplay* OR impersonat* OR storytell* OR “story-tell*”) ab(theat* OR drama* OR recit* OR “performing-art*” OR “participatory-art*” OR psychodram* OR “role-play” OR “role-plays” OR “role-playing” OR “role-player*” OR roleplay* OR impersonat* OR storytell* OR “story-tell*”) 2 OR 3 OR 4 1 AND 5
Scopus	((TITLE-ABS-KEY (suicid*)) AND (TITLE-ABS-KEY (theat* OR drama* OR recit* OR “performing-art*” OR “participatory-art*” OR psychodram* OR “role-play” OR “role-plays” OR “role-playing” OR “roleplayer*” OR roleplay* OR impersonat* OR storytell* OR “story-tell*”))) AND NOT ((INDEX (medline OR embase)) OR (PMID (1* OR 2* OR 3* OR 4* OR 5* OR 6* OR 7* OR 8* OR 9* OR 0*)))
Web of Science	TS = (suicid*) AND TS = (theat* OR drama* OR recit* OR “performing-art*” OR “participatory-art*” OR psychodram* OR “role-play” OR “role-plays” OR “role-playing” OR “role-player*” OR roleplay* OR impersonat* OR storytell* OR “story-tell*”)

**Table 2 ijerph-19-14948-t002:** Articles meeting the inclusion criteria.

Reference	Country	Study Design	Main Art Form Used	Purpose	Sample Size and Population	Period of Intervention
Bartgis et al., 2016 [82]	USA	Observational study without a control group	Role-play online	To examine the outcomes for American Indian and Alaska Native students, teachers, and faculty completing online role-play gatekeeper training simulations.	Gatekeepers-9000: university professors, college students, high school, and middle school educators	From April 2011 to December 2013
Birrane et al., 2015 [79]	Ireland	Observational study without a control group	Role play	To describe the development and evaluation of an educational intervention on youth mental health and substance misuse for primary care professionals.	Gatekeepers-30 general practitioners and other primary care professionals	Session: 2 h
Carpenter et al., 2021 [88]	USA	Observational study without a control group	Role play	To identify how to adapt the online Veteran Administration’s suicide prevention gatekeeper training program (SAVE) for community pharmacy staff.	Gatekeepers-17 community pharmacy staff members	Session: 20 min and 1 h semi-structured interview.
Coleman et al., 2015 [94]	USA	Quasi-experimental prospectively controlled study	Role play	To test hypotheses about two brief suicide prevention gatekeeper trainings (question, persuade, refer [QPR] and RESPONSE) and one longer suicide intervention skills training (Applied Suicide Intervention Skills Training [ASIST]).	Gatekeepers-126 (clinician, teachers, church leaders, coaches, corrections staff)	Session: 1 h 25 min. Period of intervention: 6 months.
Cross et al., 2007 [74]	USA	Observational study without a control group	Role play	To evaluate outcomes of a gatekeeper training for suicide prevention in a sample of non-clinicians.	Gatekeepers-76 nonclinical employees in a university hospital	Session: 1 h. Period of intervention: 6 weeks.
Cross et al., 2010 [76]	USA	Observational study without a control group	Role play	To assess and predict the impact of brief, gatekeeper training on community members’ observed skills.	Gatekeeper-50 employees at US universities	Session: 6 h. Period of intervention: 4 months.
Cross et al., 2011 [100]	USA	RCT	Role play	To compare gatekeeper training as usual with training plus brief behavioral rehearsal for school staff and parents in a school community.	Gatekeepers-114	Session: 1 h 25 min. Period of intervention: 17 months.
Fanian et al., 2015 [80]	Canada	Observational study without a control group	Multiple art forms—spoken word, sound production and design, film, photography, multimedia arts, jewellery making and visual arts.	To evaluate a creative arts workshop for Tłı ˛cho ˛ youth to explore critical community issues and find solutions together using the arts.	9 youth per day-ages ranged from 13 to 22	Session: 5 days
Fenwick et al., 2004 [92]	UK	Quasi-experimental prospectively controlled study	Role play	To evaluate the impact of two types of training courses: full day workshops with actors role-playing patients; and a half-day lecture.	Gatekeepers-107 from different disciplines	Session: 1 h 45 min. Period of intervention: 2 months.
Godoy Garraza et al., 2021 [103]	USA	RCT	Role play	To examine the impact of two training enhancements (role-play and booster) on intermediate gatekeepers training outcomes.	Gatekeepers-661 (287 QPR + Role-play; 374 QPR Alone)	Session: 1–2 h. Period of intervention: 6 months.
Goldberg et al., 2012 [97]	Russia	Observational study without a control group	Role play	To improve the education of existing primary care staff on the management of mental health disorders.	Gatekeepers-37 general practitioners, feldshers, practice nurses, psychologists, and teacher	Session: 5 days. Period of intervention: 3 months.
Gryglewicz et al., 2020 [50]	USA	Observational study without a control group	Role play	To examine the effect of role-play training on mental health practitioners’ attitudes, subjective norms, and perceived behavioral control surrounding suicide risk assessment behaviors.	Gatekeepers-137 mental health workers	Session: 4.5 h. Period of intervention: 3 years.
Høifødt et al., 2007 [73]	Norway	Observational study without a control group	Role play	To describe the experience of newly educated physicians lived experience learning processes related to treating suicidal patients.	Gatekeepers-13 medical candidates	Session: 2 days
Hutson et al., 2021 [89]	USA	Observational study without a control group	Role play-simulation	To describe objective structured clinical examinations (OSCEs) for nurse practitioner students aimed at building skills for managing a pediatric patient with acute suicidal ideation.	Gatekeepers-18 nurse practitioners, pediatric nurse practitioner and psychiatric-mental health nurse practitioners	Session: 20 min and 24 h to complete evaluation.
Kaur, 2021 [95]	India	Quasi-experimental prospectively controlled study	Role play	To investigate the effect of life skills training on the suicidal behavior of adolescents.	970 adolescents (485 experimental group; 485 control group)	Session: 40 min
Keller et al. J Soc Mark. 2017 [83]	USA	Observational study without a control group	Theater	To examine whether a community-based suicide prevention project could increase willingness to seek professional help for suicidal ideation among eastern Montana youth.	224 high school students	Session: the performance lasted 20 min, followed by a moderated 40 min Q and A session. Period of intervention: 6 months.
Keller et al., 2017 [30]	USA	Observational study without a control group	Theater	To evaluate a community-based, narrative theater project designed to increase awareness and use of suicide prevention resources among eastern Montana youth.	27 high school students and college students	Session: 12 weeks. Period of intervention: 3 years.
Keller et al., 2019 [99]	USA	Cross-sectional study	Theater	To analyze differences in Eastern Montana Caucasian and Native American youths’ experiences with stigma about mental illness that affect help-seeking for suicidal experiences.	33 high school students and college students	10 weeks
Kratz et al., 2020 [87]	USA	Observational study without a control group	Role play	To evaluate the outcomes of an educational pilot study integrating didactic instruction, readings, role-plays, and simulation for teaching suicide intervention skills.	Gatekeepers-58 Master of Social Work’s students	Session: 50 min. Period of intervention: 16 weeks.
Long et al., 2018 [101]	USA	RCT	Role-play online	To evaluate the impact of the At-Risk for Elementary School Educators online mental health role-play simulation for elementary school teachers on changes in teachers’ helping attitudes and behaviors in students experiencing psychological distress.	Gatekeepers-18,896 schoolteachers, mental health professionals	Session: 45–90 min. Period of intervention: 3 months.
Morriss et al., 1999 [72]	UK	Observational study without a control group	Role play	To devise and evaluate the retention of a new brief training package for non-psychiatrically trained multidisciplinary staff to assess suicide risk and manage suicidal patients.	Gatekeepers-33 health and voluntary workers	8 h of interview skills training (2 h sessions).
Nakagami et al., 2018 [85]	Japan	Observational study without a control group	Role play	To evaluate a suicide intervention program among medical staff.	Gatekeepers-74 medical staff members	Session: 2 h. Period of intervention: 1 month.
Nash et al., 2021 [91]	Australia	Observational study without a control group	Theater	To evaluate a workshop for health care students that used filmed vignettes from a verbatim theater play.	Gatekeepers-65 nursing, medical and allied health students and medical students only.	Session: 90 min
O’Reilly et al., 2019 [102]	Australia	RCT	Role play	To use a novel mental health first aid assessment approach involving simulated role-plays enacted by people with a lived experience of mental illness and explore participants’ and simulated patients’ views of participating in simulated role-plays of mental health crises.	Gatekeepers-22 pharmacy students	Mean duration of 28.8 min
Owen et al., 2018 [84]	USA	Observational study without a control group	Role play	To create a symposium curriculum to provide a structured, safe environment where mental health trainees and practitioners of various specialties cab obtain collegial support and education to reduce the stigma and potential isolation surrounding patient suicide.	35 mental health practitioners and trainees	Session: 4 h
Pasco et al., 2012 [93]	USA	Quasi experimental control study	Role play	To evaluate the efficacy of an experiential-based gatekeeper training, which included an emphasis on enhancing communication skills and relational connections in addition to the didactic foci of standard gatekeeper training.	Gatekeepers-65 college resident advisors	Session: 3 h
Robinson et al., 2016 [81]	Australia	Observational study without a control group	Role play	To examine the impact of delivering an evidence-based gatekeeper training package for suicide prevention (STORM^®^) in an Australian setting.	Gatekeepers-84 staff members from schools	Training package duration (2 days) and FU 8 weeks later.
Ross et al., 2021 [90]	USA	Observational study without a control group	Role play	To evaluate the Suicide Prevention for College Student Gatekeepers training program, designed to provide college students with information about the warning signs of suicide, as well as how to intervene when indicated.	Gatekeepers-65 college students	Session: 90 min. Period of intervention: 12 weeks.
Silverman et al., 2013 [77]	Canada	Observational study without a control group	Multiple art forms—performing arts (drama, music, artmaking, and writing)—arts-based approach.	To describe a two-day arts-based symposium that brought together members from diverse cultural communities.	18 members from different cultural communities including the Inuit, Mohawk, Jewish, Christian, Baha’i, South-Asian Canadian, Senior and LGBTQ communities.	Session: 2 days
Stewart et al., 2009 [75]	USA	Observational study without a control group	Multiple art forms—classes encompassing the visual, performing, musical, media, and theatre arts.	To examine the effects of the Prodigy Cultural Arts Program on at-risk and adjudicated youth in a rural and an urban locale.	350 adolescents and their parents	Session: 2 months
Thomas et al., 2006 [96]	UK (Scotland)	Cross-sectional study	Theatre	To evaluate STIGMA play and workshops, with the aim of addressing sensitive issues close to young people, such as suicide or self-harm, and to improve seeking help attitudes.	950 secondary school children	Session: 1 h and 10 min
Timmons-Mitchell et al., 2019 [86]	USA	Observational study without a control group	Role-play online	To examine the impact of a virtual training program, Kognito At-Risk role-play simulation, on the mental health and suicide prevention gatekeeping skills of middle school educators.	Gatekeepers-33,703 middle school educators	Session: 45–90 min. Period of intervention: 3 months.
Vallance et al., 2014 [78]	UK	Observational study without a control group	Role play	To develop and evaluate a novel teaching session on clinical assessment using role play simulation.	Gatekeepers-10 medical students	Session: 90 min
Wasserman et al., 2018 [98]	Estonia, Italy, Romania and Spain	Cross-sectional study	Role play	To discuss mental health in terms relevant to youth (peer support, stress, crisis, depression, suicide, and help-seeking), after their participation in the Youth Aware of Mental Health Program.	32 adolescents	Session: a five-hour program spanning three weeks.
Wasserman et al., 2015 [18]	Europe (Austria, Estonia, France, Germany, Hungary, Ireland, Italy, Romania, Slovenia, and Spain).	RCT	Role play	To report the results of the Saving and Empowering Young Lives in Europe (SEYLE) study, a largescale, multi-country, European randomized controlled trial of the school-based prevention of suicidal behavior in adolescents.	11,110 adolescents	Session: 5 h in 4 weeks. Period of intervention: 12 months.

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
