# Peer review of "Performing Arts in Suicide Prevention Strategies: A Scoping Review"

_ijerph, 2022, doi:10.3390/ijerph192214948_

Round 1

Reviewer 1 Report

I think the authors could clarify the abstract and the discussion. Such clarification would enable the authors to indicate early in the paper that most of the studies selected concerned the effect of the performing arts on the work of gatekeepers in suicide prevention work. There was less about the use of the performing arts on 'patients' or clients. Such an emphasis is fine, but it is worthwhile highlighting this. 

A detailed, well-researched study but a little more analysis of the results and what the authors identified as the elements of performing arts that significantly contributed to suicide prevention would have strengthened the concluding discussion. 

Author Response

Authors’ reply to the reviewers:

Reviewer 1

I think the authors could clarify the abstract and the discussion. Such clarification would enable the authors to indicate early in the paper that most of the studies selected concerned the effect of the performing arts on the work of gatekeepers in suicide prevention work. There was less about the use of the performing arts on 'patients' or clients. Such an emphasis is fine, but it is worthwhile highlighting this. 

We modified the abstract text in order to early clarify the reader that most of the studies selected are addressed to gatekeepers.

A detailed, well-researched study but a little more analysis of the results and what the authors identified as the elements of performing arts that significantly contributed to suicide prevention would have strengthened the concluding discussion. 

We enriched the discussion, as suggested.

Reviewer 2 Report

Thanks for inviting me to review this manuscript

This study describes art-based interventions used for suicide prevention.

I have some comments to improve this manuscript:

Introduction:

The subtitles in line 191 and line 220 in the intro section have the same numbers.

Methods:

Can you please clarify the inclusion criteria in terms of the Population (participants), the concept (intervention),etc. There is repetition when talking about the inclusion criteria. For example, in the paragraph starting from line 232 and again in the paragraph starting from line 262. I would prefer if you could add a subheading and explain the inclusion criteria in one place.

Note: Inclusion Criteria as a separate section according to JBI

Under the subheading: 2.2. Extraction and Presentation of Results: you only explain the methods of data extraction and presentation. Please move table 2 to the results section.

Results

Again repetition in the section starting from line 339 and the section starting from 426.

Please delete the repetition.

Talking about the art forms(theatre, roleplay, etc), there are supposed to be subheadings under 3.5

Overall the results need to be organised in a comparable and understandable way focusing on answering the research questions.

Author Response

Author's reply to reviewer:

Reviewer 2

Thanks for inviting me to review this manuscript

This study describes art-based interventions used for suicide prevention.

I have some comments to improve this manuscript:

Introduction:

The subtitles in line 191 and line 220 in the intro section have the same numbers.

Thank you, we changed the number (1.3 -> 1.4).

Methods:

Can you please clarify the inclusion criteria in terms of the Population (participants), the concept (intervention),etc. There is repetition when talking about the inclusion criteria. For example, in the paragraph starting from line 232 and again in the paragraph starting from line 262. I would prefer if you could add a subheading and explain the inclusion criteria in one place.

Note: Inclusion Criteria as a separate section according to JBI.

Thank you, we removed inclusion criteria from introduction and we expanded the dedicated section in the methods, with a clear reference to the PCC elements, according to which we subheaded the section.

 Under the subheading: 2.2. Extraction and Presentation of Results: you only explain the methods of data extraction and presentation. Please move table 2 to the results section.

Thank you, we moved Table 2 to the results section.

Results

Again repetition in the section starting from line 339 and the section starting from 426.

Please delete the repetition.

We deleted repetition.

Talking about the art forms (theatre, roleplay, etc), there are supposed to be subheadings under 3.5

We added subheadings 3.5.1 and 3.5.2.

Overall the results need to be organized in a comparable and understandable way focusing on answering the research questions.

We resumed in each subheading the research questions declared at the end of the introduction in order to make the text more readable and understandable.

Round 2

Reviewer 2 Report

Thank you for making the proposed changes

Can you please review the review questions under "the aim of the review section"? Please review the numbering and the sequence of questions.

There is a repetition in lines 280-282 and 285-287.

Author Response

Thank you for making the proposed changes

Can you please review the review questions under "the aim of the review section"? Please review the numbering and the sequence of questions.

Thank you for the suggestion. We reordered questions so that they match with the dedicated section in results’ section.

There is a repetition in lines 280-282 and 285-287.

We removed repetition.